# Physiological, Productive, and Soil Rhizospheric Microbiota Responses of ‘Santina’ Cherry Trees to Regulated Deficit Irrigation Applied After Harvest

**DOI:** 10.3390/plants14233611

**Published:** 2025-11-26

**Authors:** Tamara Alvear, Macarena Gerding, Richard M. Bastías, Carolina Contreras, Silvia Antileo-Mellado, Andrés Olivos, Mauricio Calderón-Orellana, Arturo Calderón-Orellana

**Affiliations:** 1Departamento de Producción Vegetal, Facultad de Agronomía, Universidad de Concepción, Chillán 3780000, Chile; talvear2019@udec.cl (T.A.); mgerding@udec.cl (M.G.); ribastias@udec.cl (R.M.B.); santileo2016@udec.cl (S.A.-M.); 2Instituto de Producción y Sanidad Vegetal, Facultad de Ciencias Agrarias y Alimentarias, Universidad Austral de Chile, Valdivia 5091000, Chile; carolina.contreras@uach.cl; 3OLIVOS Riego SpA, Curicó 3340000, Chile; andres@olivos.cl (A.O.); mcalderon@olivos.cl (M.C.-O.)

**Keywords:** *Prunus avium*, postharvest, stem water potential, water stress, plant growth-promoting bacteria, water use efficiency, ecological memory, drought resilience

## Abstract

Chile, the leading exporter of cherries (*Prunus avium* L.) in the southern hemisphere, faces sustained variations in precipitation patterns and high evaporative demand in its productive areas. The low availability of water during the period of highest environmental demand makes it essential to reduce or suspend irrigation applications. In this scenario, regulated deficit irrigation (RDI) after harvest is an efficient strategy for optimizing water use without compromising orchard yields. This study was conducted over three consecutive seasons in a traditional commercial orchard of ‘Santina’ cherry trees grafted onto Colt rootstock, evaluating the effect of two levels of RDI, moderate (MDI) and severe (SDI), on productive and ecophysiological parameters. Both treatments resulted in water savings of between 10% and 28%, without negatively affecting yield or fruit quality. The SDI treatment, despite reaching higher levels of cumulative water stress, improved intrinsic water use efficiency while maintaining stable photosynthetic efficiency. In addition, an increase in the abundance of fine roots and beneficial rhizosphere bacteria populations, such as *Azospirillum* and *Bacillus*, was observed, suggesting the activation of water resilience mechanisms mediated by plant–microbiota interaction, possibly associated with stress-induced ecological memory and microbial legacy effects. These results position after-harvest RDI as a sustainable tool for coping with climate variability and water scarcity in commercial cherry orchards.

## 1. Introduction

Chile is the leading exporter of sweet cherries (*Prunus avium* L.) in the Southern Hemisphere, accounting for more than 90% of the volume, with annual exports of 341,108 tons and approximately 70,686 hectares planted [1]. The sweet cherry cultivated area is mainly concentrated in the O’Higgins and Maule regions [2], in the central part of the country, where the most productive areas for fruit growing and wine production are located. It is characterized by a temperate Mediterranean climate with well-defined seasons: rainy winters and warm, dry summers [3]. In this area, the water demand of sweet cherry trees is estimated at between 7000 and 8000 m^3^ hectare^−1^ per season, and there are currently more than 67,000 hectares planted [2]. The demanding atmospheric conditions in central Chile, together with the high water demands of cherry trees, make water resource management and monitoring essential. In this context, the implementation of irrigation strategies becomes crucial, especially when considering the worst-case scenarios projected by climate models, which anticipate increases in ~4.1 °C in average temperature and a 39% reduction in snowpack reserves, particularly in the central zone of the country [4]. Over the past four years, the area has experienced drought conditions, with annual rainfall significantly below normal levels, showing anomalies exceeding 90% compared to historical values [5].

In Chile, under conditions of high evaporative demand and limited water availability, a common practice for cherry orchards is to reduce or suspend irrigation during the post-harvest period. This strategy saves water, allowing it to be used for other crops that are in full productive development, coinciding with the months of highest atmospheric demand. In addition, in certain cases, it is applied to induce leaf senescence through water stress, which favors the onset of endodormancy [6]. However, this practice is often carried out without physiological monitoring, which can lead to severe stress in plants.

In this context, regulated deficit irrigation (RDI) emerges as an efficient strategy for saving water during specific periods of the phenological cycle, applying irrigation volumes lower than the crop evapotranspiration (ET_c_), without compromising yield or fruit quality [7,8]. The RDI has been validated in various fruit trees, such as apricot (*Prunus armeniaca* L.) [9], Japanese plum (*Prunus salicina* L.) [10], European plum (*Prunus domestica* L.) [11], and apple (*Malus domestica* Borkh.) [12].

In sweet cherries ‘Prime Giant’, the application of RDI after harvest has resulted in water savings of up to 39%, increasing water productivity without affecting yield. However, when this strategy is applied in stages before harvest, a tendency toward reduced fruit size has been observed [13]. For this reason, deficit irrigation during fruit growth is not recommended for early-maturing varieties of various species, including cherry trees [14,15].

The success of RDI depends on precise management in terms of timing, duration, and severity. Marsal et al. [16] point out that incipient leaf wilting in cherry trees can be observed in the field when the water potential of the stem (Ψ_stem_) reaches a threshold of −1.8 MPa. Blaya-Ros et al. [17] warn that persistent stress with Ψ_stem_ values below −2.0 MPa could have significant impacts on plant physiology, even inducing collapse.

Although this irrigation strategy has been extensively studied in cherry trees, the response of different production systems to this type of management remains unknown. Different combinations of variety and rootstock respond differently to drought-induced water stress, which directly influences vegetative growth, reproduction, and nutrient absorption and transport [18].

In Chile, between 2010 and 2014, the main commercial sweet cherry cultivars were ‘Lapins’ (28.6%) and ‘Santina’ (26.3%), most commonly grafted onto the Colt rootstock [19]. Studying the response to water stress in commercial orchards established with these cultivar–rootstock combinations is particularly relevant, as these orchards, despite being considered traditional systems, remain highly productive and represent a significant portion of the national fruit-growing area. Understanding the tolerance of these orchards to water deficit is key to optimizing strategies such as RDI without compromising fruit quality or yield.

In parallel with water management, plants have various physiological mechanisms to cope with drought stress, including avoidance, escape, tolerance, and recovery [20]. Of these, tolerance and avoidance are the most relevant strategies for dealing with water deficit, as they allow the plant to resist dehydration through physiological adjustments such as stomatal closure [21].

In addition, irrigation practices induce transient changes in the soil water–oxygen balance, which alter the pH of the rhizosphere and stimulate the release of compounds that promote the proliferation of microorganisms that constitute microbiota [22]. Recent studies have highlighted the importance of these changes in the rhizosphere microbiome in modulating plant responses to abiotic stress, such as drought [23]. Communities associated with the root system, particularly plant growth-promoting rhizosphere bacteria (PGPB), establish mutualistic relationships with roots, modulating the production of phytohormones such as abscisic acid, auxins, and cytokinins, stimulating root growth, improving nutrient absorption, and activating antioxidant systems that mitigate oxidative damage [24,25,26]. One complementary strategy for improving water stress tolerance is to leverage the interactions between plant roots and soil microorganisms, particularly those of this type of bacteria. However, the composition and dynamics of these microbial communities in traditional production systems remain unexplored. Therefore, studying how water stress influences the microbial structure of the rhizosphere and root system is essential for developing strategies to improve the sustainability and productivity of fruit orchards.

To establish irrigation strategies such as RDI in productive orchards, it is necessary to understand how their physiological and microbiological mechanisms respond to maximize their benefits and move toward systems that are more resilient to drought. The objective of this study was to evaluate the effect of RDI applied after fruit harvest on plant water relations, intrinsic water use efficiency, fruit quality and yield, and the cumulative effect of water stress on the microbial composition of the rhizosphere and root system in a commercial orchard of cherry trees cv. ‘Santina’, under two levels of stress severity for three consecutive seasons.

## 2. Materials and Methods

### 2.1. Study Site

The study was conducted over three consecutive seasons (2021–2022, 2022–2023, and 2023–2024) in a commercial sweet cherry orchard (*Prunus avium* L. cv. ‘Santina’), located in Placilla (34°36′5.385″ S, 71°3′40.484″ W), Libertador General Bernardo O’Higgins Region, Chile. The orchard is located on ‘Talcarehue’ series soils, classified as Inceptisols, with a loamy-to-loamy clay texture. These soils are characterized by a well-defined structure, poorly developed horizons, and subangular blocks. They are predominantly colored in the 7.5YR group, ranging from dark brown to dark yellowish brown. The profile rests on alluvial layers composed of gravel and boulders, with a loamy–clay–sandy matrix [27]. The area has a warm Mediterranean climate, characterized by a prolonged dry season of approximately six months and a rainy winter, as classified by the CIREN agroclimatic classification [27]. The trees were planted in 2004 and grafted onto Colt rootstock (*Prunus avium* × *Prunus pseudocerasus*), trained on a central leader, with a planting frame of 2.4 m between plants and 4.5 m between rows. The orchard was irrigated using a double-line drip system, with four emitters per plant, spaced 0.4 m apart, with an emission rate of 2.2 L h^−1^ per emitter. Facing south–north, the orchard has a high-density polyethylene roofing system, which is extended after the application of cyanamide and closed after fruit set, depending on the rainfall forecast for its opening or closing. The average yield was 17 t·ha^−1^ during the 2021–2022 season and 11 t·ha^−1^ in the 2022–2023 and 2023–2024 seasons, showing a decrease in productivity in the last two years. Tree pruning was carried out manually twice during each season. The first, corresponding to winter pruning, was carried out between late May and early June, to regulate fruit load. The second pruning focused on regulating the vegetative and productive structure, favoring the entry of light into the canopy. Sweet cherries were hand-harvested, while pests, weeds, and disease management were carried out following regular commercial practices.

### 2.2. Experimental Design

The experimental design was a completely randomized block design, with four replicates and repeated measurements over three consecutive seasons. Three irrigation treatments were applied immediately after fruit harvest, starting on the same day as harvest and maintained during the post-harvest period (November to March) (Table 1). The control treatment consisted of conventional irrigation practices designed to meet 100% of crop evapotranspiration (ET_c_) (Figure 1), calculated based on local climatic conditions and the orchard’s phenological state. In contrast, Regulated Deficit Irrigation (RDI) treatments involved temporarily suspending the water supply until specific thresholds of stem water potential (Ψ_stem_) were reached. Considering a stress threshold of Ψ_stem_ at −1.3 MPa [28], two levels of deficit were established: in the moderate RDI treatment (MDI), irrigation was interrupted until Ψ_stem_ reached values between −1.3 and −1.6 MPa, at which point irrigation was reestablished. On the other hand, in the severe RDI treatment (SDI), watering was suspended until Ψ_stem_ dropped to values between −1.6 and −2.0 MPa. Irrigation cuts for treatments during the post-harvest period were carried out using plastic shut-off valves installed at the beginning of the drip irrigation lines corresponding to the RDI treatments. This configuration allowed manual control of water flow, based on monitoring Ψ_stem_, resuming irrigation only when values reached the defined water stress thresholds.

Each experimental unit consisted of 12 plants distributed in three adjacent rows, corresponding to a block-treatment combination. Measurements were taken on the two central plants in the middle row, while the side rows and remaining plants were considered isolation edges.

Before harvest, commercial irrigation practices were applied uniformly throughout the orchard, regardless of the treatment assigned. Water requirements were estimated based on the calculation of crop evapotranspiration (ET_c_ = ET_o_ × K_c_). The crop coefficients (K_c_) used in this study, from flowering (September) to leaf fall (early May), were obtained from FAO Manual 56 [30] and were based on conventional practices for the sweet cherry fruit tree.

In order to estimate the ET_c_ values for the orchard, satellite image analysis was used to compare maximum irrigation requirements with cumulative irrigation during the season. The ET_c_ estimate for the center pivot irrigation system was made using the SPIDERwebGIS^®^ platform (Participatory Information System, Decision Support and Expert Knowledge for Water Management in Irrigated Basins). This tool was developed within the framework of the European PLEIADES project [31] and is currently managed by AgriSat Iberia, Albacete, Spain (http://www.spiderwebgis.org/ and https://www.agrisat.es/en) (accessed on 25 May 2025). In addition, SPIDERwebGIS^®^ integrates the Satellite Agricultural Platform (PLAS), developed by the Agricultural Research Institute (INIA) [32,33], which allows the crop coefficient (K_c_) to be determined based on the crop’s Normalized Difference Vegetation Index (NDVI).

### 2.3. Environmental Conditions

Data on global solar radiation (W m^−2^), precipitation (mm), reference evapotranspiration (m^3^ ha^−1^), relative humidity (%), and air temperature (°C) were obtained from an automatic weather station (InstaWeather model, InstaCrops, Santiago, Chile), located 148 m from the experimental plot. The average values of the main climate variables during the three seasons evaluated are presented in Appendix A. This station provided daily records of climatic variables, from which reference evapotranspiration (ET_o_) was calculated using the Penman–Monteith model proposed by FAO56 [30]. Precipitation values were subsequently adjusted to account for effective precipitation delivered to the plant, following the methodology proposed by the FAO [34]. Vapor pressure deficit (VPD) was calculated according to the equation described by Howell and Dusek [35].(1)$${\mathrm{VPD}}_{\mathrm{air}\;}=0.6108\;\cdot \;\exp\left(\frac{17.27\;\cdot \;T}{T+237.3}\right)\cdot \;1-\left(\frac{\mathrm{RH}}{100}\right)$$
where RH is air relative humidity and T is air temperature.

The volume of water applied in the trial was measured by installing eight volumetric meters (Dishnon, Arad Ltd., Dalia, Israel), distributed so that one meter was assigned to each deficit irrigation treatment. Each experimental block had two meters, which were located at the beginning of each irrigation line corresponding to the RDI treatments. The devices were installed on 19 November 2021.

The photosynthetically active photon flux density (PPFD, µmol m^−2^ s^−1^) and leaf area index (LAI) were measured weekly using a ceptometer (LP-80, Decagon Instruments, Washington, DC, USA). Each plant sampled was evaluated at midday, with four replicates per plant. Internal PPFD measurements were taken 0.2 m below the canopy, recording values at distances of 0.05, 0.20, 0.30, and 0.40 m from the trunk. External PPFD was estimated in the inter-row, 1.75 m away from the plant and 1.6 m above ground level.

### 2.4. Plant Water Status, Physiology, and Growth

Weekly, during the tree post-harvest period (November to March), the Ψ_stem_ was measured in one leaf per plant sampled, corresponding to the central plant used for physiological measurements. The selected leaves were shaded, located in the lower third of the plant, and mature and healthy, with no visual symptoms of biotic or abiotic stress. Measurements were taken at midday, between 12:00 and 15:00, using a pressure chamber (model PMS-615, PMS Instruments, Portland, OR, USA). Before measurement, the leaves were covered with an opaque, aluminized, airtight bag for at least 40 min, following the methodology described by McCutchan and Shackel [36].

Stomatal conductance (mmol m^−2^ s^−1^) was measured simultaneously with Ψ_stem_, using a compact portable porometer (LI-600, LI-COR, Lincoln, NE, USA) on three mature leaves exposed to the sun, located in the apical third of the shoots. Photosystem II (PSII) efficiency was evaluated as *Fv*/*Fm* using a chlorophyll fluorescence meter (Pocket PEA, Hansatech Instruments, Norfolk, UK). For this determination, the leaves were dark-adapted for 30 min using leaf clips, as described by Reyes-Díaz et al. [37]. Subsequently, the minimum fluorescence (*F*_0_) and maximum fluorescence (*Fm*) values were recorded, from which the variable fluorescence (*Fv*) was calculated as the difference between *Fm* and *F*_0_. Finally, PSII efficiency was obtained using the following relationship:(2)$$\frac{Fv}{Fm}\;=\;\frac{Fm\;-\;F_{0}}{Fm}$$

### 2.5. Water Stress Integral

To estimate cumulative water stress, the water stress integral (S_Ψ_) was calculated, adapting the model proposed by Myers [38]. This model consists of the sum of the Ψ_stem_ recorded during the evaluation period, which in this case corresponds to the post-harvest period. However, instead of using a fixed reference value for water potential, the Ψ_stem_ was estimated based on the expected vapor pressure deficit (VPD) under environmental conditions, as reported by McCutchan and Shackel [36] for plum trees.

The estimate was made based on measurements taken at regular intervals of n days, corresponding to weekly assessments, over a total period of approximately 90 days per season. In the 2021–2022 season, 15 measurements were taken, as in the 2022–2023 season, while in the 2023–2024 season, 11 measurements were taken. This methodology allowed the integral of accumulated water stress (S_Ψ_) to be calculated using the following formula:(3)$$\mathrm{WSI}\;=\;\sum_{i\;=\;1}^{n}(\Psi_{\mathrm{stem}\;i,i+1}\;-\;\Psi_{\mathrm{base}})\cdot \;n$$
where Ψ_stem i,i+1_ corresponds to the average value of the stem water potential for any time interval between two consecutive measurements (i, i+1); Ψ_base_ represents the reference value of the expected water potential under the specific environmental conditions of each interval, estimated based on the VPD; and n corresponds to the number of days included in each measurement interval.

The cumulative was S_Ψ_ (S_Ψ1_ + S_Ψ2_ + S_Ψ3_) obtained by summing the seasonal integrals, providing a quantitative measure of long-term water stress exposure.

### 2.6. Intrinsic Water Use Efficiency

During the post-harvest period of the last season (1st week of March 2024), five young leaves and five mature leaves were selected per treatment, ensuring that they were healthy, fully expanded, and from the middle zone of the central plants in each experimental block. Sampling was conducted at 10:00 a.m., a time representative of high photosynthetic activity, to estimate intrinsic water use efficiency (WUE_i_) through differences in the carbon isotope ratio (δ^13^C), as proposed by Bchir et al. [39].

The leaf samples were dried separately in an oven at 70 °C for 48 h until they reached a constant weight. They were then ground and savvy to obtain a fine, homogeneous powder. The carbon isotope ratio (δ^13^C) was determined using an elemental analysis system coupled with isotope ratio mass spectrometry (EA-GSL, Sercon, UK; 20–22 IRMS, Sercon, UK). Before each analysis, an ultra-high-grade carbon dioxide reference gas (Ultra High-Grade CO_2_, Linde Group) was injected to correct for possible deviations. A calibrated laboratory standard (Corn Flour SCC2256, Sercon, UK) was included every ten samples as an internal quality control.

The carbon isotopic composition of each sample was calculated using the following equation, according to Brugnoli and Farquhar [40] and Farquhar et al. [41]:(4)δC13 ‰=C13C12sample−C13C12standardC13C12standard · 1000
where ^13^C/^12^C_sample_ and ^13^C/^12^C_standard_ are the measured ^13^C/^12^C ratios for the leaf sample and the PDB standard (Pee Dee Belemnite), respectively.

### 2.7. Yield Components and Fruit Quality

During the 2022–2023 and 2023–2024 seasons, at harvest time (Table 1), fruit load (kg plant^−1^) and total number of fruits were estimated on two plants evaluated per block-treatment combination. Harvest time was determined according to the commercial management criteria of the orchard, using a color chart as a reference. Fruit load was measured by manual harvesting, using a platform scale. In the same evaluation, a sample of 50 sweet cherries per plant was randomly selected to analyze fruit quality.

The individual fresh weight of each fruit (g) was determined using a precision scale with an accuracy of ±0.1 g (APTP457A, Electronic Scale Balance, Kuala Lumpur, Malaysia).

The color of each fruit was determined using a portable colorimeter (CR-10, Konica Minolta, Tokyo, Japan), using the CIELAB color space, which provides the coordinates L*, a*, and b*, where L* represents lightness, a* the red/green axis, and b* the yellow/blue axis. The concentration of soluble solids (°Brix) in the manually extracted juice was measured with a digital refractometer (HI 96801, Hanna Instruments, Woonsocket, RI, USA), previously calibrated with distilled water before each measurement. Additionally, fruit condition and quality defects were assessed, including the presence of double fruits, cracking (classified by fruit zone), deep suture, and other visible abnormalities.

### 2.8. Root System Characterization

At the end of the post-harvest period of the last season, and once the corresponding evaluations had been completed, the number and size of roots were determined in all irrigation treatments (n = 3). The evaluations were conducted using 1 m-deep and 1 m-wide pits located in the central row of the experimental blocks.

Each pit was divided into a grid of 100 subunits of 0.1 m^2^, allowing for a detailed evaluation of the root system. Before counting and classification, the exposed profile of the pit was meticulously cleaned with an agricultural knife to expose the roots present.

The roots were classified according to their diameter into the following categories: fine (<0.5 mm), thin (0.5–2.0 mm), medium (2.0–5.0 mm), medium to thick (5.0–7.0 mm), and thick (>7.0 mm). For each pit, the proportion of roots in each category was determined in relation to the total number of roots observed.

### 2.9. Evaluations of the Cultivable Soil Microbiota

During the final season of evaluations, the root system was characterized by extracting samples of free soil and rhizospheric soil from the effective rooting zone (0.4–0.6 m depth) using a disinfected drill. Samples were collected from three of the four evaluated blocks, obtaining 250 g of free soil and 250 g of rhizospheric soil per block. All tools were disinfected with 70% ethanol before sampling. The samples were placed in labeled polyethylene bags and stored in a refrigerated container at 5 °C until laboratory analysis. In the laboratory, each sample was homogenized, and subsamples of 10 g (free soil) and 1 g (rhizospheric soil) were suspended in 90 mL of sterile saline solution (0.801% NaCl) in Erlenmeyer flasks. These suspensions were agitated in an orbital shaker at 150 rpm for two hours. Serial dilutions ranging from 10^−2^ to 10^−5^ were prepared and used for microbial quantification.

To assess microbial populations, different inoculation volumes were used depending on the target group: 100 µL aliquots per plate for *Azospirillum* spp., *Azotobacter* spp., and actinobacteria, and 20 µL microdroplets per plate for mesophilic aerobic and facultative anaerobic bacteria, strict anaerobes, *Pseudomonas* spp., *Bacillus* spp., phosphate-solubilizing bacteria, and nitrogen-fixing bacteria. All culture media were prepared with deionized water and sterilized in an autoclave at 120 °C and 1 atm for 20 min. Media were poured into Petri dishes under aseptic conditions in a laminar flow chamber. Microbial colonies were counted using the microdroplet technique. The inoculum was distributed on the agar surface with a sterile glass rod. Plates were incubated at 25 ± 2 °C in darkness for three days, except for Congo Red medium, which was incubated for seven days. Colony-forming units (CFU) were counted from dilutions yielding between 30 and 300 colonies.

Specific media used included Jensen agar (2 g L^−1^ dextrose, 0.2 g L^−1^ casein, 0.5 g L^−1^ K_2_HPO_4_, 0.2 g L^−1^ MgSO_4_·7H_2_O, traces of FeCl_3_·6H_2_O, 2.5% agar) for actinobacteria; LG medium for *Azotobacter* spp. [42]; RC medium for *Azospirillum* spp. [43]; PCA medium for mesophilic aerobic and facultative anaerobic bacteria; pK medium for phosphate-solubilizing bacteria; Burk medium for nitrogen-fixing bacteria; nutrient agar (MERCK) for *Bacillus* spp. and strict anaerobes (incubated in GasPak™ EZ Anaerobe Container System Sachets); and King B medium for *Pseudomonas* spp. Additionally, a 50 g subsample from each experimental unit was dried in a forced-air oven at 60 °C until constant weight to determine soil dry matter content.

### 2.10. Statistical Analysis

The data were subjected to analysis of variance (ANOVA) after verifying the assumptions of normality in distribution (Shapiro–Wilk test), homogeneity of variances (Levene’s test), and additivity (Tukey’s test). Differences between means were determined using the LSD multiple comparison test (α = 0.05). The relationship between physiological and microbiological variables was determined using cubic and quadratic regressions. All statistical analyses were performed using RStudio software version 2025.05.0 (RStudio, Posit, Boston, MA, USA).

## 3. Results

### 3.1. Environmental Conditions and Characterization of Irrigation

During the seasons evaluated (September to May), seasonal variations in environmental conditions were observed (Figure 2), with the three seasons coinciding with the highest air temperatures in December, January, and February, and with the lowest relative humidity values during those same months (Figure 2A–C). These conditions were associated with higher atmospheric evaporative demand during this period. In terms of accumulated precipitation, the first two seasons showed a similar pattern, with no rainfall events recorded during the evaluation period (November to March). In contrast, the third season (Figure 2C) had higher precipitation rates, with rainfall events in October, November, and February, reaching accumulations of up to 40 mm in November.

Seasonal differences were observed in the monthly crop evapotranspiration (ET_c_) during the three evaluated seasons (Figure 3B). In all seasons, ET_c_ increased sharply from November to December, coinciding with the onset of active shoot growth. December consistently marked the period of highest water demand, with ET_c_ values reaching approximately 3.0 mm in the first season (2021–2022), increasing to over 3.7 mm in the second season (2022–2023), and around 3.2 mm in the third season (2023–2024). This trend continued into January, maintaining elevated ET_c_ levels across all seasons.

Regarding the crop coefficient (K_c_) (Figure 3A), maximum values were reached in December at all three stations, with values close to 0.8, followed by a progressive decrease towards the end of the cycle. These seasonal variations in ET_c_ and K_c_ were consistent with the patterns of vapor pressure deficit (VPD), which showed the highest monthly averages during December, January, and February in all seasons (Appendix A). However, VPD values under peak demand conditions were reached in February (~3.6 kPa) during the second season (2022–2023), compared to approximately 3.0 kPa in the first season. In contrast, during the last season, the month with the highest demand was January (~3.6 kPa). The high VPD during the summer months reflects the combined effect of high temperatures and low relative humidity (Appendix A).

The cumulative ET_c_ showed seasonal variability (Table 2). In the first season (2021–2022), ET_c_ was the lowest, reaching 6403 m^3^ ha^−1^. In the second season (2022–2023), ET_c_ increased significantly to 7889 m^3^ ha^−1^, representing the highest value recorded. In the third season (2023–2024), ET_c_ slightly decreased to 7138 m^3^ ha^−1^. Notably, the third season also recorded the highest effective precipitation, with a total of 1552 m^3^ ha^−1^.

Regarding irrigation practices, RDI treatments consistently reduced the volume of water applied compared to the control treatment across all seasons. In the first season, water applications under MDI and SDI were 8.2% and 11.7% below the seasonal ET_c_, respectively. The second season showed the most pronounced reductions, with SDI decreasing water application by 28.3% and MDI by 19.3%, corresponding to 15.9% and 25.2% below ET_c_. In the third season, water savings were lower, with reductions of 9.9% (MDI) and 11.3% (SDI), and irrigation volumes were close to 100% of ET_c_. In contrast, the control treatment consistently applied water volumes close to or above crop requirements, exceeding ET_c_ by 11% in the third season.

Ψ_stem_ remained below the baseline threshold (~−0.6 MPa) from harvest (stage 87) to the onset of leaf senescence (stage 92) in all treatments and seasons (Figure 4). Although all plants experienced water stress after harvest, each season showed a distinct pattern in terms of occurrence, duration, and severity. Control treatment consistently maintained the highest Ψ_stem_ values, which generally ranged from −0.8 to −1.2 MPa during the first two seasons (2022 and 2023). In both seasons, control plants reached the minimum value (~−1.2 MPa) only once. In contrast, during the third season (2024), the control treatment recorded three instances in which Ψ_stem_ dropped between −1.0 and −1.2 MPa, associated with commercial irrigation management practices.

Following irrigation cuts, both deficit treatments quickly reached their defined stress thresholds. In the first season (2022), the most severe stress occurred for approximately four weeks, from late January to early March. During this period, MDI plants reached minimum Ψ_stem_ values close to −1.6 MPa, while SDI plants dropped to −2.0 MPa. The second season (2023) showed a more pronounced and prolonged decrease in Ψ_stem_, with more severe stress lasting approximately six weeks, from December to the end of February. In this season, MDI plants reached minimum values of −1.7 MPa, and SDI plants consistently recorded the lowest values, with minimums between −1.8 and −2.0 MPa.

In the third season (2024), however, the treatments did not differ significantly during most of the season in terms of water stress, except for a measurement taken during the second week after stage 92, in which SDI recorded a lower Ψ_stem_ than the MDI and control treatments.

### 3.2. Water Stress Integral

Post-harvest water stress accumulation (S_Ψ_) varied according to the treatment applied (Figure 5), being systematically lower in the control and higher in the RDI treatments. In the 2022 and 2023 seasons, MDI and SDI showed significantly more negative accumulations than the control, with differences greater than −10 MPa and −25 MPa, respectively, at the end of the period. Water stress intensified in both 2022 and 2023 after stage 91, although in the second season, the RDI treatments (MDI and SDI) were already significantly different from the control immediately after harvest. In both seasons, the SDI treatment reached accumulations below −50 MPa at the end of the monitoring period, while MDI showed an intermediate level of stress in all seasons. Both treatments showed a marked increase in accumulation after stage 91. Although in 2024 the SΨ levels were similar between treatments, the accumulation pattern remained: Control < MDI < SDI, however, without significant differences, reaching accumulation values close to each other.

The RDI strategies influenced total accumulated water stress (Figure 6). The control treatment showed the lowest stress accumulation, with a value of −80 MPa. In contrast, the MDI and SDI treatments reached significantly more negative accumulations, with values close to −115 MPa and −140 MPa, respectively. Although both RDI treatments accumulated higher stress levels than the control, they did not differ from each other.

### 3.3. Physiological Responses

The application of post-harvest irrigation treatments resulted in a significant decrease in stomatal conductance (g_s_) in plants subjected to RDI compared to the control treatment during the three seasons evaluated (Figure 7). Plants under commercial irrigation consistently maintained the highest g_s_ values, with maximums close to 500 mmol m^−2^ s^−1^ in 2022, approximately 280 mmol m^−2^ s^−1^ in 2023, and above 300 mmol m^−2^ s^−1^ in 2024, while the minimum values did not fall below 150 mmol m^−2^ s^−1^. In contrast, the RDI treatments showed lower values, with differences from the control that were evident at different times depending on the season. In 2022, these differences were observed from the fourth week after the start of Stage 91; in 2023, from the third week after harvest; and in 2024, the control treatment showed significantly higher values for three consecutive weeks from the fifth week after harvest. The MDI treatment reached minimum g_s_ values of less than 200 mmol m^−2^ s^−1^ in 2022, which fell to just over 100 mmol m^−2^ s^−1^ in 2023. For its part, the SDI treatment recorded the lowest levels of stomatal conductance consistently in all seasons, with minimums close to 200 mmol m^−2^ s^−1^ in 2022 and values close to 50 mmol m^−2^ s^−1^ in 2023 and 2024.

Regression analysis between Ψ_stem_ and g_s_, based on data collected during the three seasons evaluated, showed a positive relationship (R^2^ = 0.94, *p* < 0.001) (Figure 8). Maximum stomatal conductance was associated with Ψ_stem_ values close to −0.7 MPa, maintaining a relative conductance between 80% and 100%. As Ψ_stem_ decreased to −1.4 MPa, g_s_ was reduced by approximately 40%, a stress level mainly reached by RDI treatments. When the plants reached a Ψ_stem_ of −2.0 MPa, a threshold consistently recorded in the SDI treatment, g_s_ was reduced by more than 60%, with relative values below 40%. On the other hand, even under Ψ_stem_ conditions above −0.7 MPa, a slight decrease in g_s_ of close to 10% was observed when Ψ_stem_ reached −0.5 MPa.

No significant differences were observed in photosystem II efficiency (*Fv*/*Fm*) or LAI between post-harvest irrigation treatments during the three seasons evaluated (Table 3). The average *Fv*/*Fm* values remained stable between treatments, ranging between 0.65 and 0.76, while LAI values fluctuated between 5.47 and 6.47 without showing differences.

### 3.4. Intrinsic Water Use

The levels of carbon 13 (^13^C) discrimination were evaluated after the application of post-harvest irrigation treatments during three consecutive seasons. The results showed significant differences between treatments (Figure 9). The SDI treatment showed significantly less negative ^13^C discrimination values, compared to the Control and MDI treatments.

### 3.5. Reproductive and Vegetative Growth Responses

The application of RDI treatments after harvest did not significantly affect most reproductive parameters. No differences were observed between treatments (*p* > 0.05) in orchard yield (Table 4), yield per plant, or crop load during the 2023 and 2024 seasons. Similarly, most fruit quality attributes, including color, diameter, and soluble solids, showed no significant differences between treatments (Table 5). The only exception was observed in fruit weight during the 2023 season, where the MDI treatment (11.7 g) produced significantly heavier fruits than the control treatment (9.9 g); however, this effect was not consistent, as it was not observed in the following season.

No significant differences in fruit quality or condition parameters were observed among irrigation treatments during the first evaluated season (2023). However, during the second season (2024), apical cracking was detected across all treatments, with mean values ranging from 21.3% to 27.5% (Table 5). Although this physiological disorder was present, it did not appear to be associated with any specific irrigation treatment.

The relative abundance of fine roots was higher in the MDI and SDI treatments compared to the control, with an increase of more than 0.1 units in both cases, following the application of treatments for three post-harvest seasons (See Figure 10).

### 3.6. Cultivable Microbiota Response

No significant differences were found between irrigation treatments in free soil bacterial populations. However, in the rhizosphere, RDI treatments (MDI and SDI) showed significant effects on cultivable microbiota (Figure 11). The populations of aerobic and facultative anaerobic mesophiles (AER), *Azospirillum* spp. (AZO), and Actinobacteria (ACT) were significantly higher in the MDI and SDI treatments compared to the control. In addition, AER were significantly more abundant in SDI treatment than in MDI. A similar trend was observed in nitrogen-fixing bacteria (NFB) and *Bacillus* spp. (BAC), where both deficit treatments had significantly higher populations than the control. In the case of *Azotobacter* spp. (AZOT), SDI treatment showed a significantly higher population than control, while MDI showed no differences from the other treatments. No significant differences were detected between treatments in the populations of anaerobic mesophiles (ANA), *Pseudomonas* spp. (PSE), or phosphate-solubilizing bacteria (PSB).

Regression analysis showed a positive relationship between accumulated water stress values (−S_Ψ_) during the three seasons and PGPB bacterial populations in the rhizosphere, observed in the RDI treatments (See Figure 12). As cumulative water stress increased, that is, as S_Ψ_ values became more negative, an increase was observed in the populations of aerobic and facultative anaerobic mesophiles (AER), nitrogen-fixing bacteria (NFB), *Azotobacter* spp. (AZOT), and *Bacillus* spp. (BAC). The coefficients of determination obtained were R^2^ = 0.76 (*p* < 0.01) for AER, R^2^ = 0.77 (*p* < 0.01) for NFB, R^2^ = 0.64 (*p* < 0.01) for AZOT, and R^2^ = 0.52 (*p* < 0.05) for BAC.

## 4. Discussion

The results of this study show that traditional commercial orchards of ‘Santina’ sweet cherry trees established in central Chile can be irrigated at a deficit after harvest for more than two consecutive seasons without generating carryover effects on production parameters. The implementation of this strategy resulted in water savings ranging from 10% to 28% compared to commercial irrigation (Table 2).

The period of regulated deficit irrigation (RDI) application coincided with a time of year characterized by (1) maximum atmospheric evaporative demand (Figure 3B), (2) limited availability of irrigation water, and (3) the coexistence of other fruit crops in phenological stages more sensitive to water stress, whose demand must be prioritized to avoid negative impacts on yield and quality. In this context, the application of RDI in ‘Santina’ sweet cherry trees represents an efficient strategy for redistributing water resources to more sensitive species or cultivars during this period. The application of RDI is particularly relevant since ‘Santina’ is an early-maturing cultivar compared to other commercial cultivars [44], in which harvest operations in Chile are carried out in late November (Table 1), when the evaporative demand has not yet reached its annual maximum.

Before the application of RDI treatments, plants exhibited stem water potentials (Ψ_stem_) slightly below the baseline (Figure 4), suggesting near-optimal water conditions. This reflects commercial irrigation strategies aimed at balancing water supply to avoid both overwatering and moderate stress near harvest. However, postharvest irrigation is often reduced or suspended in cherry orchards, especially under high evaporative demand and limited water availability. The control treatment in this study confirmed this practice, showing consistently sub-baseline Ψstem values throughout the experimental period, with minimum values near −1.2 MPa (Figure 4), indicating moderate water stress even under standard irrigation.

Once the RDI treatments were applied, Ψ_stem_ exhibited high variability throughout each season. From harvest to stage 91, Seasons 1 and 3 showed comparable declines in Ψ_stem_ across all treatments, reaching moderate water stress levels (Ψ_stem_ close to −1.3 MPa) by the end of this period. These progressive declines were associated with increases in maximum daily vapor pressure deficit (VPD) and solar radiation between December and January (Appendix A). During the first half of the experimental period, differences in Ψ_stem_ among treatments in both seasons were small and inconsistent.

In Season 2, the gap between Ψ_stem_ values of the control and RDI treatments was the largest observed during the study, with minimum stomatal conductance values in RDI plants reaching only half of those recorded in control plants (Figure 7). These significant reductions in stomatal conductance under RDI during Season 2 may have limited photosynthesis and affected the normal growth and development of vegetative organs. It has been reported that stomatal conductance (g_s_) is highly dependent on Ψ_stem_ [28,36], which is consistent with the results of the present study, where gs responded to changes in water status across all three seasons (Figure 7). A strong correlation (R^2^ = 0.94) was observed, establishing that at a Ψ_stem_ threshold of −1.8 MPa, plants closed their stomata by approximately 60% as a physiological mechanism to regulate transpiration and prevent irreversible dehydration damage [45]. Conversely, under over-irrigation conditions (Ψ_stem_ ~ −0.5 MPa), a reduction in stomatal conductance of around 10% was detected, possibly linked to decreased root oxygenation [46]. Despite these stress events and partial stomatal closures in the RDI treatments, the photochemical efficiency of photosystem II (*Fv*/*Fm*) remained stable throughout all seasons (Table 3), with values within the expected range under normal growth conditions [47]. Since stage 91 marks the end of shoot growth, it would be reasonable to expect reduced vegetative development in RDI-treated plants compared to controls. However, leaf area index (LAI) measurements at the end of each growing season showed no significant differences among irrigation treatments (Table 3). Sweet cherry trees typically reach their maximum shoot growth rates from late spring to early summer, with peak elongation occurring between late November and early December in the Southern Hemisphere [48]. Therefore, the results suggest that up to four one-week episodes of severe water stress between harvest and the end of shoot growth (December to January), a period when shoot elongation rate is already declining, did not negatively impact vegetative development.

While Season 3 exhibited slight variations in Ψ_stem_ among irrigation treatments between stages 91 and 92, Seasons 1 and 2 showed two and three instances, respectively, of severe water stress in SDI plants (Ψ_stem_ < −1.5 MPa). In MDI plants, water stress severity ranged from moderate to severe during the first and second seasons, with two to three one-week stress events occurring before the leaves began to fade in color. The greater severity and cumulative duration of water stress in Season 2 are evident when analyzing the seasonal cumulative water stress integral (S_Ψ_). During this season, RDI treatments exhibited higher cumulative water stress levels compared to the control treatment (Figure 5).

The prolonged exposure of SDI and MDI plants to severe water stress may have potentially disrupted the normal progression of flower development, as floral differentiation for the following season occurs between stages 89 and 92. However, results clearly showed that yield per tree was not affected by irrigation treatments in any of the three seasons (Table 4), suggesting that flower differentiation was likely not impaired by postharvest water stress. Furthermore, the significantly higher values S_Ψ_ observed in RDI treatments did not impact fruit quality in the subsequent season (Table 5).

These findings are consistent with previous studies, which reported that postharvest water deficits with Ψ_leaf_ values between −1.6 and −1.9 MPa allowed irrigation to be reduced by 20–25% over four consecutive seasons without negatively affecting yield or fruit quality [49]. Similarly, Blanco et al. [50] found that, in the ‘Prime Giant’ cultivar, deficit irrigation equivalent to 55% of ET_c_ during the postharvest period did not affect fruit size or flowering, even after three consecutive seasons.

In contrast to the findings of this study, several previous reports have indicated that post-harvest water deficits can negatively affect vegetative development and yield in the following season [7], mainly due to reduced carbohydrate reserves, which are essential for early shoot growth, flowering, and fruiting [51,52,53]. Marsal et al. [16] reported that Ψ_stem_ values below −1.5 MPa may compromise yield, given the close relationship between sugar concentration in buds and vegetative growth [54]. Similarly, Blanco et al. [13] documented a ~7% reduction in canopy volume after three seasons of RDI with Ψ_stem_ < −1.6 MPa. However, no adverse physiological effects attributable to RDI treatments were observed in the present study, suggesting that the intensity and duration of the applied stress were not sufficient to trigger such responses. On the other hand, the average soluble solids content (°Brix) remained below the optimal commercial threshold of 16 °Brix across all seasons, and although fruit diameters exceeded the pre-sizing limit (>22 mm), most fruits were classified within the smaller commercial categories (L and XL). Additionally, the higher yield observed in the first season relative to the second may be explained by interannual yield fluctuations, rather than by differences in irrigation management. A slight incidence of apical cracking was also observed in the second season, attributed to cuticular water absorption following November rainfall events (Figure 2), which affected all treatments similarly. Overall, these results suggest that the occurrence of cracking, as well as the observed variations in yield and fruit quality parameters, were more closely related to environmental and management conditions than to the applied post-harvest water deficits.

The water savings achieved through RDI strategies, combined with the moderate to severe stress levels and their controlled duration, together with the efficient stomatal regulation observed particularly in the SDI treatment, contributed to physiological adjustments that enhanced water use efficiency. The significant differences observed in δ^13^C isotopic discrimination indicate that the postharvest water deficit effectively modified intrinsic water use efficiency (WUE_i_). The less negative δ^13^C values recorded in SDI (−27.8 ‰) reflect reduced stomatal conductance accompanied by increased carbon assimilation and transpiration, consistent with the g_s_ reductions observed across the three seasons (Figure 9). These results suggest that under moderate to severe stress conditions, plants maintained photosynthetic activity while limiting water loss, leading to improved instantaneous WUE_i_ [39,41]. This physiological response is further supported by the stability of photochemical efficiency (*Fv*/*Fm*) (Table 3).

When plants reached moderate to severe levels of water stress, the RDI treatments were irrigated in the same way as the control, maintaining Ψ_stem_ between approximately −0.8 and −1.2 MPa. However, irrigation cuts generated water fluctuations due to soil drying and rehydration, which may have caused changes in the microbiological dynamics between treatments (Figure 11). The differences found between treatments in the rhizosphere bacterial populations in this study are consistent with those reported by Meisner et al. [55], who noted that drying–rewetting cycles can induce microbial legacy effects, modifying bacterial communities compared to conditions of constant soil moisture [56].

In this study, the RDI treatments induced changes in the bacterial population composition in the rhizosphere, even after three consecutive seasons (Figure 11). This effect was more evident in SDI, where irrigation cuts were longer at certain times and the lowest values of Ψ_stem_ and S_Ψ_ were recorded. This condition would have favored the proliferation of aerobic bacteria, which increased by 21% in MDI and 27% in SDI compared to the control, probably due to greater soil drying and greater oxygen diffusion in the rhizosphere.

In contrast to the recent results of Calderón-Orellana et al. [11], in plum trees, where three times fewer anaerobic bacteria were observed under water deficit conditions compared to over-irrigation management, the RDI treatments in the present study showed no significant differences in this bacterial group compared to the control. The lack of significance suggests that the control was not exposed to over-irrigation conditions, remaining constantly below the soil moisture baseline (Figure 4) and close to 100% ET_c_ (Table 2), which would have prevented the generation of hypoxia conditions favorable for the development of these bacteria.

Canarini et al. [57] proposed that repeated exposure to droughts can induce an “ecological memory” characterized by greater functional resilience of bacterial communities. Under water stress, plants increase the synthesis of abscisic acid (ABA), modifying the amount and composition of root exudates, which act as chemical signals and carbon sources that promote certain beneficial microbial colonization [58,59].

In this sense, the increase in plant growth-promoting bacteria (PGPB) observed in this study could be due to both the alteration of root exudates and changes in soil oxygenation. A significant increase in *Bacillus*, *Azospirillum*, nitrogen-fixing bacteria, Actinobacteria, and *Azotobacter* was observed, even after three seasons with highly variable plant water status and environmental conditions (i.e., summer rainfall (<7 mm) and VPD; Figure 2). Most of these bacteria are strictly aerobic, except some facultative anaerobic Actinobacteria, which could explain their higher numbers in the RDI treatments.

The populations of *Azotobacter* spp., nitrogen-fixing bacteria, *Bacillus* spp., and aerobic bacteria increased by more than 20% compared to the control, showing a linear relationship with S_Ψ_ accumulated over the three seasons (Figure 12), reinforcing the hypothesis of stress-induced ecological memory. This interaction is key, as PGPBs contribute to adaptation to water deficit by regulating water uptake, modifying root architecture, and synthesizing phytohormones such as indoleacetic acid (IAA) and gibberellins [60]. Strains within *Azotobacter* spp., for example, can produce cytokinins that promote moisture retention, while there have been reports on *Bacillus* spp. and *Azospirillum* spp. as being able to synthesize IAA and gibberellins that promote root growth [61,62]. This coincides with the observed increase in the abundance of fine roots (Figure 10), which accounted for about 70% of the roots sampled and were stimulated by more than 30% by the RDI treatments. This is extremely relevant in orchards grafted onto ‘Colt’, characterized by a shallow root system and low drought tolerance [63].

These results suggest that applying RDI in orchards of ‘Santina’ cherry trees grafted onto more vigorous rootstocks after harvest, in ranges of Ψ_stem_ between −1.6 and −2.0 MPa, allows for water savings of up to 28% without compromising vegetative growth, yield, or fruit quality, even under moderate to severe stress events. Despite the high variability in the effects of irrigation treatments on Ψ_stem_ from one year to the next and the subtle differences between treatments observed last season, isotopic and microbiological analyses showed positive, consistent impacts of RDI on intrinsic water use efficiency and beneficial rhizosphere microbiota. The stimulation of beneficial bacterial communities and the increase in fine root abundance suggest strengthened water resilience mechanisms and greater resource use efficiency. Taken together, these findings reinforce the idea that the post-harvest RDI strategy optimizes water use, preserves physiological integrity, maintains productivity, and improves plant interaction with beneficial microbiota. This ensures the sustainability of commercial sweet cherry orchards in the face of water scarcity and climate variability.

## 5. Conclusions

This study demonstrates that regulated deficit irrigation applied after harvest in traditional ‘Santina’ sweet cherry trees grafted onto Colt rootstock can be implemented over several seasons without compromising vegetative growth, yield, or fruit quality. Water savings ranged from 10% to 28%, and the levels of water stress achieved did not affect the main production parameters or photosynthetic efficiency. Severe deficit irrigation improved intrinsic water use efficiency and stimulated the proliferation of beneficial rhizosphere bacteria, including *Azospirillum*, *Bacillus*, and nitrogen-fixing bacteria, suggesting the activation of water resilience mechanisms and possible ecological memory effects. The increase in fine root abundance supports improved water absorption and soil oxygenation under stress conditions. These findings position post-harvest RDI as a sustainable and adaptive strategy for cherry production in Mediterranean climates, promoting water efficiency and orchard resilience to climate variability.

## Figures and Tables

**Figure 1 plants-14-03611-f001:**
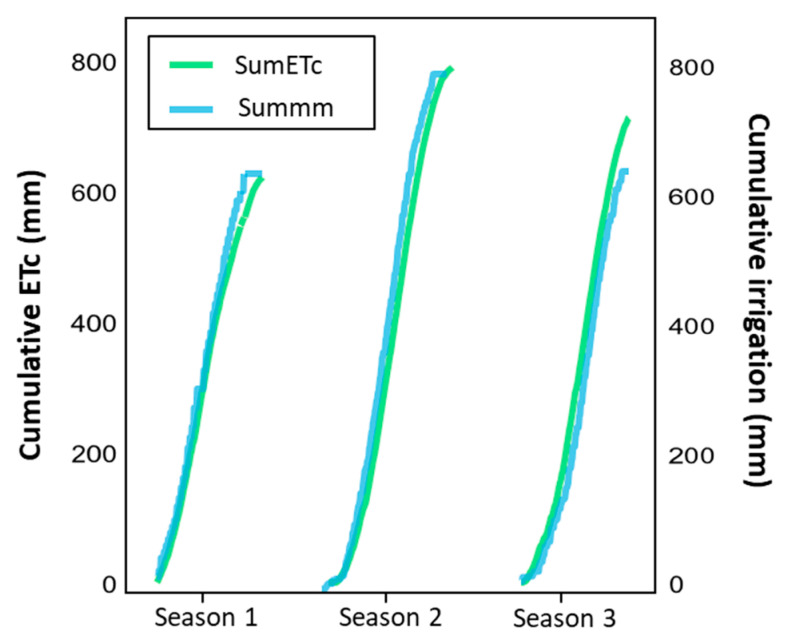
Applied irrigation (Summm) in the ‘Control’ treatment versus crop evapotranspiration (ET_c_) (SumETc) over three seasons in Placilla, O’Higgins Region, Chile: Season 1 (2021–2022), Season 2 (2022–2023), and Season 3 (2023–2024).

**Figure 2 plants-14-03611-f002:**
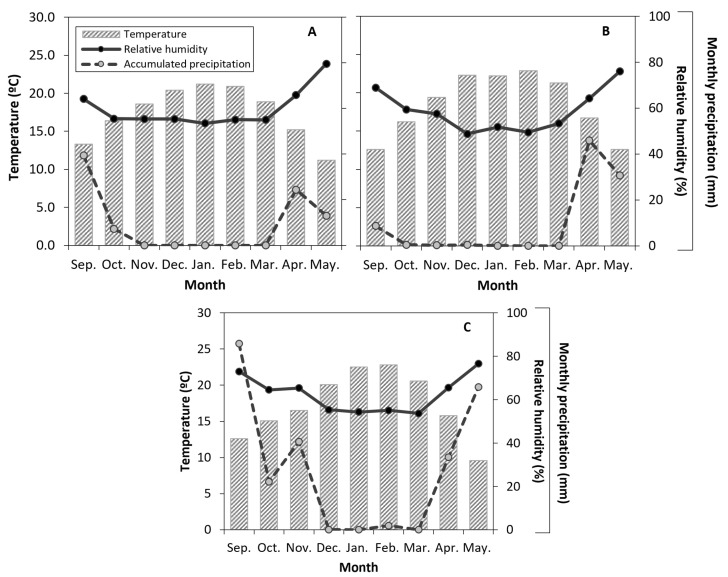
Monthly temperature, relative humidity, and accumulated precipitation recorded at the nearest weather station in a sweet cherry orchard in Placilla, O’Higgins Region, from anthesis to the onset of leaf senescence (September to May) during the 2021–2022 (Season 2022, (**A**)), 2022–2023 (Season 2023, (**B**)), and 2023–2024 (Season 2024, (**C**)) seasons.

**Figure 3 plants-14-03611-f003:**
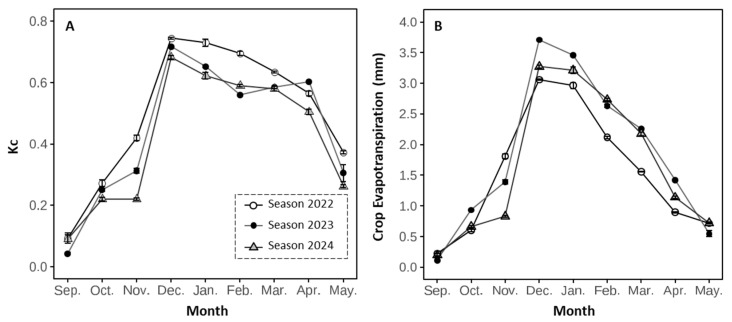
Monthly average of (**A**) crop coefficient (Kc) and (**B**) crop evapotranspiration in a ‘Santina’ sweet cherry orchard in Placilla, O’Higgins Region, from anthesis to the onset of leaf senescence (September to May) during the 2021–2022 (Season 2022), 2022–2023 (Season 2023), and 2023–2024 (Season 2024) seasons. Error bars represent ±1 se based on four experimental replications (n = 4).

**Figure 4 plants-14-03611-f004:**
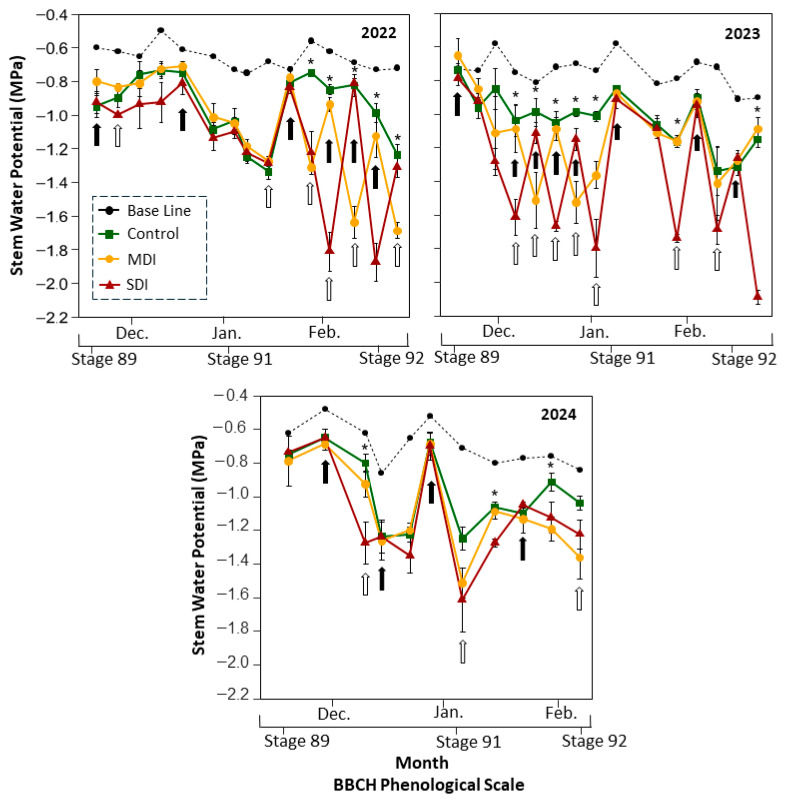
Stem water potential at midday (12:00–15:00 h) in a sweet cherry orchard under three irrigation treatments: Control (commercial irrigation), MDI (moderate regulated deficit irrigation after harvest), and SDI (severe regulated deficit irrigation after harvest), conducted in Placilla, O’Higgins Region, during the 2022, 2023, and 2024 seasons. Black arrows indicate the irrigation cutoff dates, and white arrows indicate the resumption of irrigation in RDI treatments (MDI and SDI). The black line represents the baseline for optimal water status. Asterisks (*) indicate significant differences (*p* ≤ 0.05, n = 4). Phenological stages are shown according to the BBCH scale: Stage 89 (fruit ripe for harvest), Stage 91 (shoot growth completed; foliage green), and Stage 92 (Leaves begin to fade color) [29].

**Figure 5 plants-14-03611-f005:**
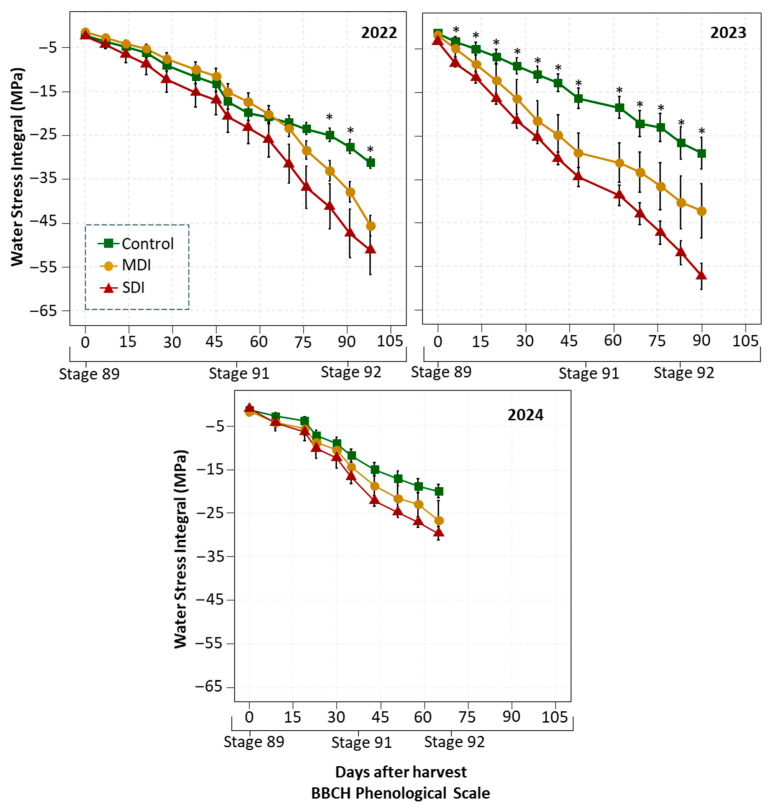
Water stress integral (MPa) after harvest in a sweet cherry orchard under three irrigation treatments: Control (commercial irrigation), MDI (moderate regulated deficit irrigation after harvest), and SDI (severe regulated deficit irrigation after harvest), conducted in Placilla, O’Higgins Region, during the 2022, 2023, and 2024 seasons. Asterisks (*) denote significant differences (*p* ≤ 0.05, n = 4). Phenological stages are shown according to the BBCH scale: Stage 89 (fruit ripe for harvest), Stage 91 (shoot growth completed; foliage green), and Stage 92 (Leaves begin to fade color) [29].

**Figure 6 plants-14-03611-f006:**
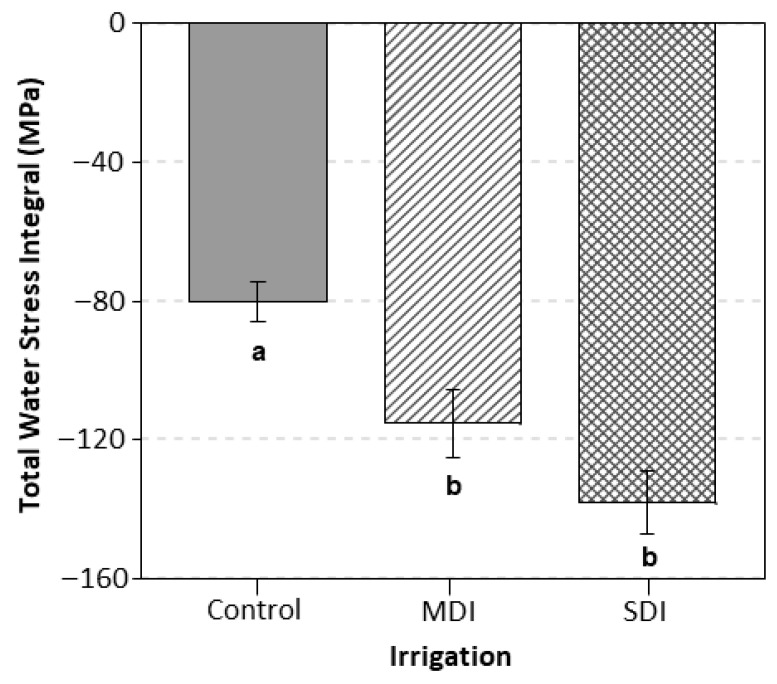
Cumulative water stress integral (MPa) after harvest in a sweet cherry orchard under three irrigation treatments: Control (commercial irrigation), MDI (moderate regulated deficit irrigation after harvest), and SDI (severe regulated deficit irrigation after harvest), conducted in Placilla, O’Higgins Region, during the 2022, 2023, and 2024 seasons. Different letters indicate significant differences between treatments according to LSD Fisher’s test (*p* ≤ 0.05, n = 4).

**Figure 7 plants-14-03611-f007:**
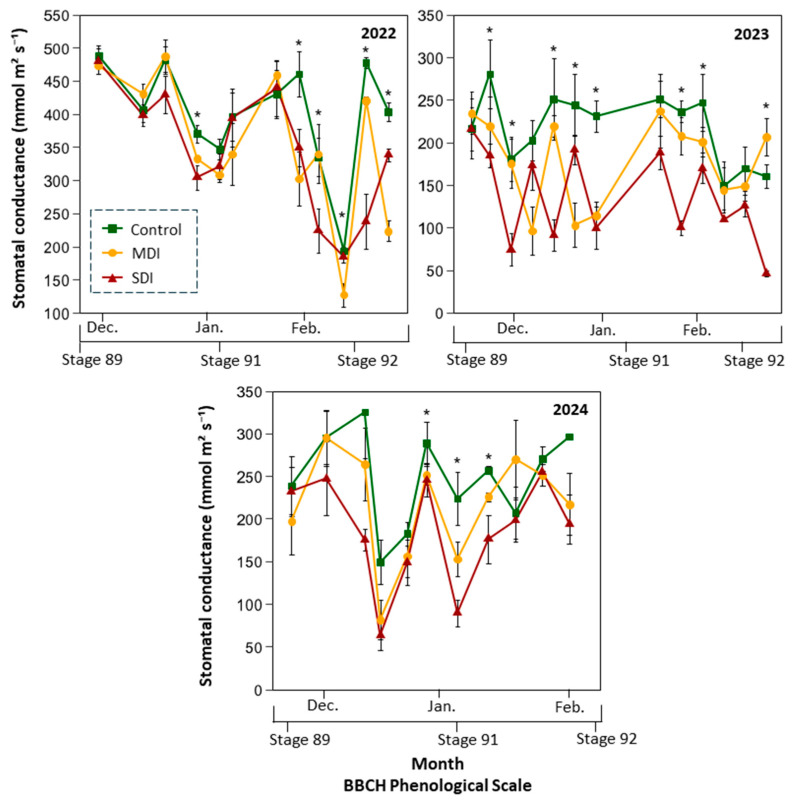
Stomatal conductance (mmol m^−2^ s^−1^) at midday (12:00–15:00 h) in a sweet cherry orchard under three irrigation treatments: Control (commercial irrigation), MDI (moderate regulated deficit irrigation after harvest), and SDI (severe regulated deficit irrigation after harvest), conducted in Placilla, O’Higgins Region, during the 2022, 2023, and 2024 seasons. Asterisks (*) denote significant differences (*p* ≤ 0.05, n = 4). Phenological stages are shown according to the BBCH scale: Stage 89 (fruit ripe for harvest), Stage 91 (shoot growth completed; foliage green), and Stage 92 (Leaves begin to fade color) [29].

**Figure 8 plants-14-03611-f008:**
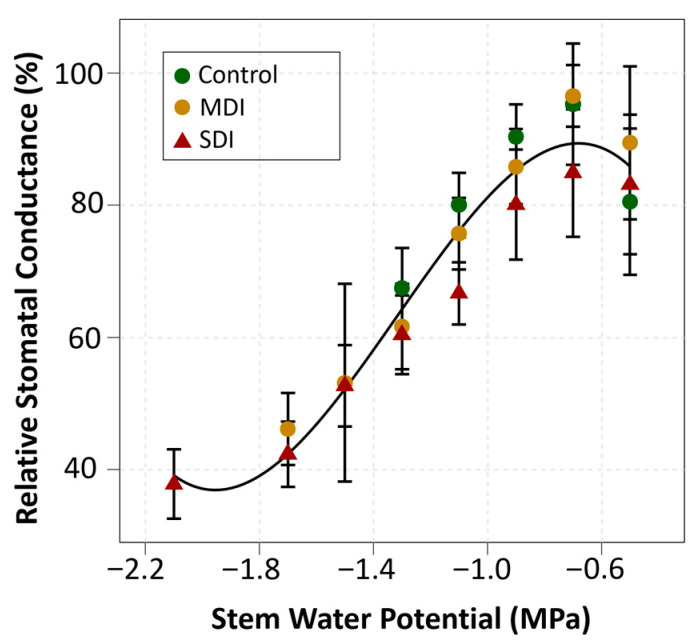
Relationship between midday stem water potential (12:00–3:00 p.m.) and relative stomatal conductance (%) in cherry trees under three irrigation treatments: Control (commercial irrigation), MDI (moderate post-harvest deficit), and SDI (severe post-harvest deficit) in a commercial orchard located in Placilla, O’Higgins Region. The data corresponds to the average for the 2022, 2023, and 2024 seasons. The adjusted polynomial regression (y = 28,349 − 202,643x − 200,637x^2^ − 50,749x^3^) was significant (*p* < 0.001, n = 3), with a coefficient of determination R^2^ = 0.94.

**Figure 9 plants-14-03611-f009:**
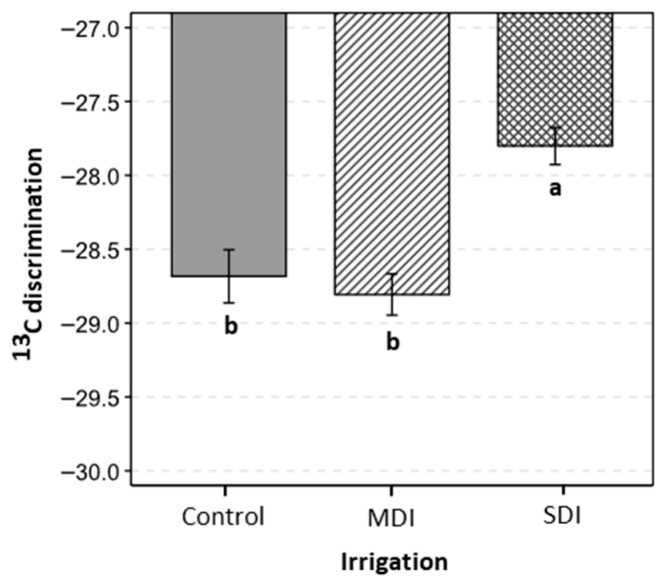
^13^C discrimination in cherry trees (1st week of March 2024) under three irrigation treatments: Control (commercial irrigation), MDI (moderate deficit irrigation after harvest), and SDI (severe deficit irrigation after harvest), in Placilla, O’Higgins Region, during the 2022, 2023, and 2024 seasons. Different letters indicate significant differences between treatments according to LSD Fisher’s test (*p* ≤ 0.05, n = 4). Error bars represent ±1 es.

**Figure 10 plants-14-03611-f010:**
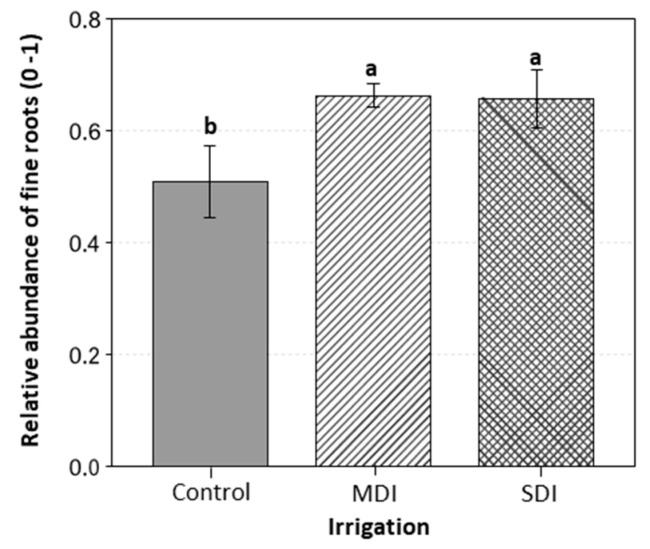
Relative abundance of fine roots in ‘Santina’ cherry trees (n = 3). Evaluated in ‘Santina’ cherry trees subjected to three post-harvest irrigation treatments: Control (commercial irrigation), MDI (moderate deficit), and SDI (severe deficit), in a commercial orchard located in Placilla, O’Higgins Region. Error bars represent ±1 es. Different letters indicate significant differences between treatments according to Fisher’s LSD test (*p* < 0.05).

**Figure 11 plants-14-03611-f011:**
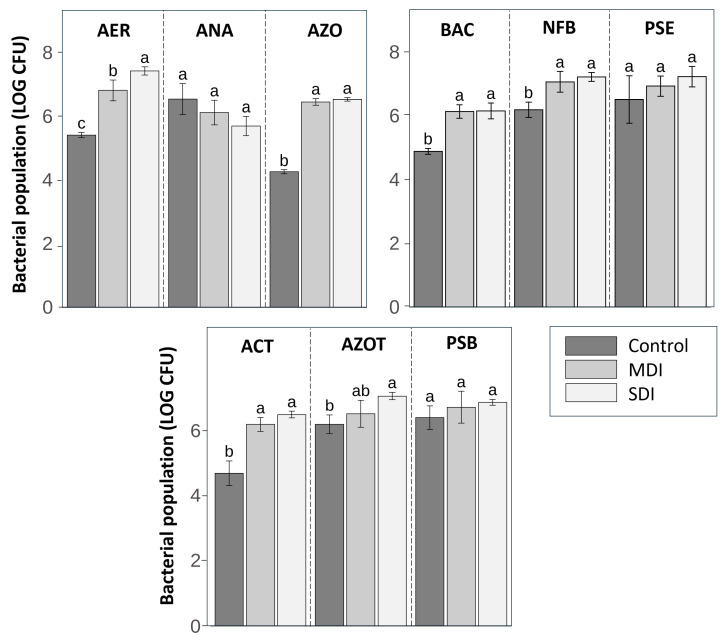
Bacterial population (LOG CFU) in the rhizosphere of ‘Santina’ cherry trees (n = 3). AER: aerobic and facultative anaerobic mesophiles; ANA: anaerobic mesophiles; AZO: *Azospirillum* spp.; BAC: *Bacillus* spp.; NFB: nitrogen-fixing bacteria; PSE: *Pseudomonas* spp.; ACT: Actinobacteria spp.; AZOT: *Azotobacter* spp.; PSB: phosphate-solubilizing bacteria. Evaluated in ‘Santina’ cherry trees subjected to three post-harvest irrigation treatments: Control (commercial irrigation), MDI (moderate deficit), and SDI (severe deficit), in a commercial orchard located in Placilla, O’Higgins Region. Different letters indicate significant differences between treatments according to Fisher’s LSD test (*p* < 0.05, n = 3). Error bars represent ±1 es.

**Figure 12 plants-14-03611-f012:**
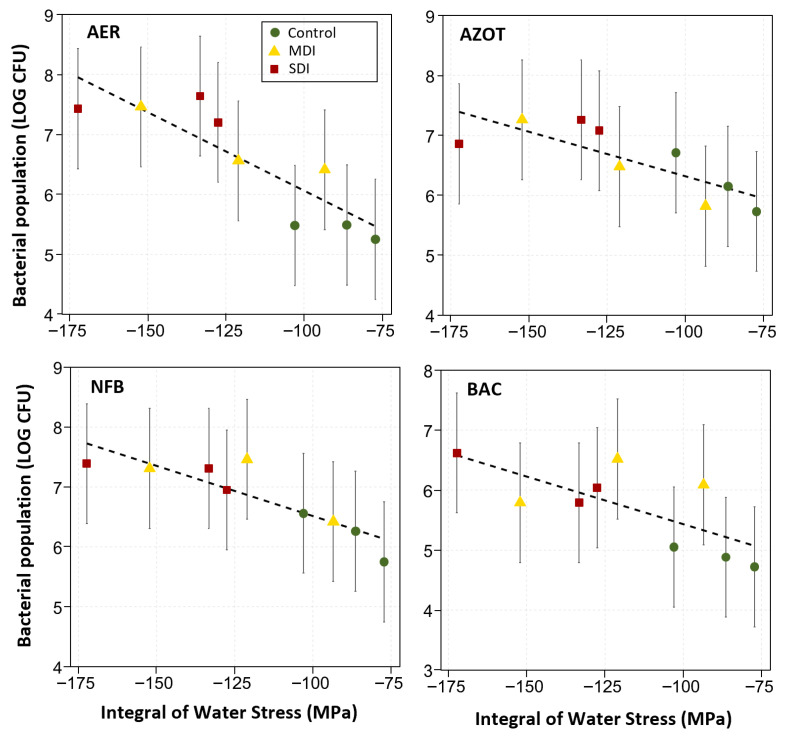
Relationship between bacterial population (LOG CFU) in the rhizosphere of ‘Santina’ cherry trees and cumulative water stress (MPa) during three post-harvest seasons (n = 3). The bacterial groups included are: AER (aerobic mesophiles and facultative anaerobes), AZOT (*Azotobacter* spp.), NFB (nitrogen-fixing bacteria), and BAC (*Bacillus* spp.). Evaluated in ‘Santina’ cherry trees subjected to three post-harvest irrigation treatments: Control (commercial irrigation), MDI (moderate deficit), and SDI (severe deficit), in a commercial orchard located in Placilla, O’Higgins Region. Error bars represent the model prediction at 68% (±1 standard deviation). The relationships were adjusted using linear regression equations: AER: y = 3.45 − 0.03x (R^2^ = 0.76, *p* < 0.01); AZOT: y = 4.83 − 0.001x (R^2^ = 0.64, *p* < 0.01); NFB: y = 4.83 − 0.02x (R^2^ = 0.77, *p* < 0.01); BAC: y = 3.84 − 0.02x (R^2^ = 0.52, *p* < 0.05).

**Table 1 plants-14-03611-t001:** Date of occurrences of several phenological stages of ‘Santina’ sweet cherries from an orchard in Placilla, O’Higgins Region, Chile, during the 2021–2022, 2022–2023, and 2023–2024 seasons.

Phenological Stages *	Season		
	2021–2022	2022–2023	2023–2024
Stage 65	September 12th	September 10th	September 9th
Stage 67	September 26th	September 27th	September 29th
Stage 81	October 25th	October 28th	October 31st
Stage 89	November 19th	November 17th	November 23rd
Stage 95	May 1st	May 1st	May 1st

* The date of occurrence for each phenological stage was recorded when the characteristic visual traits of that stage were observed in at least 50% of the evaluated plants. The stages were defined as follows: Stage 65—Full flowering; Stage 67—Flower fading; Stage 81—Beginning of fruit coloring; Stage 89—Fruit ripe for harvest; and Stage 95—50% of leaves discolored or fallen [29].

**Table 2 plants-14-03611-t002:** Descriptive analysis of cumulative values of irrigation, effective precipitation, and applied water (irrigation + precipitation), and difference in applied water (%) under three postharvest irrigation treatments: Control (commercial irrigation), MDI (moderate regulated deficit irrigation after harvest), and SDI (severe regulated deficit irrigation after harvest), conducted in a ‘Santina’ sweet cherry orchard located in Placilla, O’Higgins Region, during the 2022 (2021–2022), 2023 (2022–2023), and 2024 (2023–2024) seasons. Effective precipitation was estimated using the FAO [33] methodology.

Cumulative Values	Irrigation Treatment		
	Control	MDI	SDI
**Season 2022**			
Crop evaporation (m^3^ ha^−1^)	6403		
Irrigation (m^3^ ha^−1^)	6474	5425	5197
Effective Precipitation (m^3^ ha^−1^)	454		
Applied water (m^3^ ha^−1^)	6928	5879	5651
Difference in Applied Water (%)	-	−15.1%	−18.4%
**Season 2023**			
Crop evaporation (m^3^ ha^−1^)	7889		
Irrigation (m^3^ ha^−1^)	7972	6388	5648
Effective Precipitation (m^3^ ha^−1^)	250		
Applied water (m^3^ ha^−1^)	8222	6638	5898
Difference in Applied Water (%)	-	−19.3%	−28.3%
**Season 2024**			
Crop evaporation (m^3^ ha^−1^)	7138		
Irrigation (m^3^ ha^−1^)	6370	5583	5477
Precipitation (m^3^ ha^−1^)	1552		
Applied water (m^3^ ha^−1^)	7922	7135	7029
Difference in Applied Water (%)	-	−9.9%	−11.3%

**Table 3 plants-14-03611-t003:** Seasonal Photosystem II Efficiency (*Fv*/*Fm*) and Final Leaf Area Index (LAI) under Three Postharvest Irrigation Treatments, Control (Commercial Irrigation), MDI (Moderate Regulated Deficit Irrigation after Harvest), and SDI (Severe Regulated Deficit Irrigation after Harvest), in a ‘Santina’ Sweet Cherry Orchard Located in Placilla, O’Higgins Region, during the 2022 (2021–2022), 2023 (2022–2023), and 2024 (2023–2024) Seasons. Values are means ± standard deviation (SD); ns indicates no significant differences according to the ANOVA test (*p* > 0.05, n = 4).

Photosystem II Efficiency	Irrigation Treatment			
Season	Control	MDI	SDI	*p*-Value
2021–2022	0.67 ± 0.14	0.66 ± 0.14	0.65 ± 0.18	ns
2022–2023	0.68 ± 0.14	0.69 ± 0.10	0.69 ± 0.11	ns
2023–2024	0.76 ± 0.07	0.75 ± 0.09	0.73 ± 0.14	ns
**Leaf Area Index**				
2021–2022	5.70 ± 1.48	5.98 ± 0.78	5.47 ± 1.27	ns
2022–2023	5.67 ± 1.24	5.97 ± 1.08	5.67 ± 1.49	ns
2023–2024	6.47 ± 0.98	5.93 ± 0.85	5.52 ± 0.26	ns

**Table 4 plants-14-03611-t004:** Yield parameters of ‘Santina’ sweet cherry trees under three postharvest irrigation treatments: Control (commercial irrigation), MDI (moderate deficit irrigation after harvest), and SDI (severe deficit irrigation after harvest), conducted in a commercial orchard located in Placilla, O’Higgins Region, during the 2023 (2022–2023) and 2024 (2023–2024) seasons. “ns” indicates non-significant differences.

	Irrigation Treatment			
	Control	MDI	SDI	*p*-Value
**Season 2023**				
Orchard yield (Mg ha^−1^)	22.6 ± 2.5	19.8 ± 3.9	21.5 ± 2.7	ns
Plant yield (kg tree^−1^)	24.5 ± 2.7	20.6 ± 5.4	23.3 ± 3.0	ns
Crop load (fruits tree^−1^)	2472 ± 268	1825 ± 360	2261 ± 287	ns
**Season 2024**				
Orchard yield (Mg ha^−1^)	14.6 ± 2.7	12.4 ± 2.5	15.0 ± 2.9	ns
Plant yield (kg tree^−1^)	15.8 ± 2.9	13.4 ± 2.7	16.2 ± 3.1	ns
Crop load (fruits tree^−1^)	1435 ± 263	1183 ± 242	1571 ± 303	ns

**Table 5 plants-14-03611-t005:** Fruit quality parameters of ‘Santina’ sweet cherry trees under three postharvest irrigation treatments: Control (commercial irrigation), MDI (moderate deficit irrigation after harvest), and SDI (severe deficit irrigation after harvest), conducted in a commercial orchard located in Placilla, O’Higgins Region, during the 2023 (2022–2023) and 2024 (2023–2024) seasons. Different letters indicate significant differences between treatments according to LSD Fisher’s test (*p* ≤ 0.05, n = 4). “ns” indicates non-significant differences.

	Irrigation Treatment			
	Control	MDI	SDI	*p*-Value
**Season 2023**				
Color				
C*	24.4 ± 1.0	23.7 ± 6.9	24.7 ± 5.5	ns
h°	12.0 ± 0.7	11.7 ± 2.0	12.2 ± 1.8	ns
Weight (g)	9.9 b ± 1.0	11.7 a ± 1.0	10.3 ab ± 0.5	*p* ≤ 0.05
Polar diameter (mm)	24.5 ± 0.5	25.6 ± 1.7	24.6 ± 1.4	ns
Equatorial diameter (mm)	25.7 ± 0.9	27.2 ± 2.2	26.0 ± 1.8	ns
Soluble solids (°Brix)	14.3 ± 0.2	15.0 ± 1.6	15.0 ± 0.9	ns
**Season 2024**				
Color				
C*	24.5 ± 3.6	25.9 ± 6.4	28.6 ± 11.5	ns
h°	13.8 ± 1.1	13.9 ± 1.0	14.3 ± 2.3	ns
Weight (g)	11.0 ± 0.9	11.3 ± 0.9	10.3 ± 0.3	ns
Polar diameter (mm)	25.9 ± 1.0	25.8 ± 0.4	24.5 ± 0.6	ns
Equatorial diameter (mm)	27.6 ± 1.3	27.6 ± 0.9	26.4 ± 0.4	ns
Soluble solids (°Brix)	15.4 ± 0.8	15.9 ± 1.5	15.4 ± 1.7	ns
Apical cracking (%)	21.6% ± 2.3%	27.5% ± 2.5%	21.3% ± 8.9%	ns

C* represents chroma and h° represents hue angle, both derived from the CIELAB color space variables L, a*, and b*.

## Data Availability

The original contributions presented in this study are included in the article and Appendix A. Further inquiries can be directed to the corresponding author.

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
