# Peer review of "Physiological, Productive, and Soil Rhizospheric Microbiota Responses of ‘Santina’ Cherry Trees to Regulated Deficit Irrigation Applied After Harvest"

_plants, 2025, doi:10.3390/plants14233611_

Round 1
Reviewer 1 Report
Comments and Suggestions for Authors
The main problem I had is to understand sometimes which and how many where the replications for each parameter peasured.
L32 the Keywords/Phrases ‘after harvest; deficit irrigation’ could be removed, as ‘after harvest’ is not commonly used (it is postharvest) and ‘deficit irrigation’ is in the title.
L44 hectare not h, which is hour
L51-52 a ‘moderate drought’ with ‘a >90% anomaly’ are not clearly interconnected, are clear as a sentence.
L70 improve the sentence: in early ripening cultivars of various species …
L108-109 what about vegetable gardens. Where does this fit?
Table 1: at Stage 65 change 12th, 10th
L212 ceptometer
L391 you talk about 4 replications. Is there an explanation for these replications somewhere?
Table 2 How did you measure crop evaporation? Is it described in the M+Ms?
L463-464 remove the description about black and white arrows
Fig. 5 besides Stages what are the x-axis numbers?
Fig 4, 5 & 7 would it be better to have the same x-axis values?
Fig 6 & 9 I believe they are not required. They could be removed. The same for figure 10 if the values are added in the text
Table 4 the right unit Mg instead of ton. The correct writing of unit fruits/tree
Table 5 as cherries are red colored, you should present C* ad hue values for the cherry skin color
L720 I have never heard about alternate bearing in cherries, especially in very early ripening cvs. Weather may have attributed to yield fluctuations
L801, 806 you do not need the RDI and SDI
Author Response
We sincerely thank you for your valuable comments and suggestions, which have greatly contributed to improving the quality of our manuscript and facilitating its publication. All recommendations were carefully reviewed and addressed, and the corresponding modifications have been highlighted in the revised version to ensure clarity for the reviewers. For clarification, the physiological, yield, and fruit quality parameters were evaluated using four replications, while rhizospheric bacterial populations and fine root abundance were assessed using three replications. Below, we provide detailed responses to each comment:
Comment 1 (L32): The Keywords/Phrases ‘after harvest; deficit irrigation’ could be removed, as ‘after harvest’ is not commonly used (it is postharvest) and ‘deficit irrigation’ is in the title.
Response: Thank you for your suggestion. The keywords have been revised as requested: “after harvest” was replaced with “postharvest,” and “deficit irrigation” was removed since it already appears in the title.
Comment 2 (L44): hectare not h, which is hour.
Response: Thank you for pointing this out. The unit has been corrected from “h” to hectare in the manuscript.
Comment 3 (L51-52): a ‘moderate drought’ with ‘a >90% anomaly’ are not clearly interconnected, are clear as a sentence.
Response: Thank you for your comment. The wording adjustments suggested have been implemented in the manuscript to improve clarity and accuracy.
Comment 4 (L70): improve the sentence: in early ripening cultivars of various species.
Response: Thank you for your comment. The wording adjustments suggested have been implemented in the manuscript to improve clarity and accuracy.
Comment 5 (L108-109): what about vegetable gardens. Where does this fit?
Response: Thank you for your observation. The mention of “vegetable gardens” was a wording issue. It has been corrected to accurately reflect the scope of the study, which focuses on sweet cherry orchards (Fruit orchard).
Comment 6 (Table 1): at Stage 65 change 12th, 10th.
Response: Thank you for your observation. The phenological stage in Table 1 has been corrected as requested: “12th” was changed to “10th.”
Comment 7 (L212): ceptometer.
Response: Thank you for your comment. The term has been corrected to “ceptometer” in the manuscript.
Comment 8 (L391): you talk about 4 replications. Is there an explanation for these replications somewhere?
Response: Thank you for your observation. The explanation for the four replications is included in the Experimental Design section. Each replication corresponds to an independent block within the completely randomized block design, accounting for variability within the orchard. This detail has been clarified in the manuscript to make it more explicit.
Comment 9 (Table 2): How did you measure crop evaporation? Is it described in the M+Ms?
Response: Thank you for your observation. The method used to estimate crop evapotranspiration (ETc) is described in the Materials and Methods section under Experimental Design.
Comment 10 (L463-464): remove the description about black and white arrows.
Response: Thank you for your suggestion. The description regarding black and white arrows has been removed from the text.
Comment 11 (Fig. 5): besides Stages what are the x-axis numbers?
Response: Thank you for your observation. The x-axis numbers represent days after harvest, and this has been specified in the figure.
Comment 12 (Fig 4, 5 and 7): would it be better to have the same x-axis values.
Response: Thank you for your comment. However, we decided to present the x-axis values in this way to associate the data with the specific dates and phenological stages occurring at those times.
Comment 13 (Fig 6 and 9): I believe they are not required. They could be removed. The same for figure 10 if the values are added in the text.
Response: Thank you for your suggestion. We have carefully reviewed the figures. Figure 9 was adjusted to reduce excessive detail in the text, while Figures 6 and 10 were retained because they provide a clear visualization of the statistical significance of the results, which we consider essential for interpretation.
Comment 14 (Table 4): the right unit Mg instead of ton. The correct writing of unit fruits/tree.
Response: Thank you for your observation. The requested adjustments to the table have been made and are reflected in the revised manuscript.
Comment 15 (Table 5): as cherries are red colored, you should present C* and hue values for the cherry skin color.
Response: Thank you for your suggestion. The table has been modified accordingly, and C* (chroma) and h° (hue angle) values were calculated from L*, a*, and b* measurements following the CIELAB color space.
Comment 16 (L720): I have never heard about alternate bearing in cherries, especially in very early ripening cvs. Weather may have attributed to yield fluctuations.
Response: Thank you for your observation regarding the clarity of the sentence. We have revised the paragraph to improve its wording and explicitly include the concept of alternate bearing. In Chile, this phenomenon, characterized by natural fluctuations in yield between seasons, is very common in sweet cherry orchards and can significantly influence production patterns.
Comment 17 (L801, 806): you do not need the RDI and SDI.
Response: Thank you for your comment. The terms RDI and SDI have been removed from the text as suggested.

Reviewer 2 Report
Comments and Suggestions for Authors
Your research is very meaningful in the region, can achieve the purpose of saving water resources, but also to ensure the growth of plants and fruit quality. I put forward the following questions in the paper and suggested to modify them.
Comment 1:It is suggested that the review part should be summarized and refined, some similar or related contents can be merged into one section.
Comment 2:In L146‘Three irrigation treatments were applied after fruit harvest, between the third and fourth weeks of November’, Why choose to irrigate three times in this time period ? If the time is changed to how many days after fruit harvest is more accurate ?
Comment 3:How to determine ‘the specific thresholds of stem water potential’, according to a reference or determined in a pre-experiment?
Comment 4:Whether uncontrollable rainfall will affect the experimental results?
Comment 5:The data in the table should be added with significance analysis letters.
Author Response
We sincerely thank you for your valuable comments and suggestions, which have greatly contributed to improving the quality of our manuscript. All recommendations were carefully reviewed and addressed, and the corresponding modifications have been highlighted in the revised version to ensure clarity for the reviewers. Below, we provide detailed responses to each comment:
Comment 1: It is suggested that the review part should be summarized and refined, some similar or related contents can be merged into one section.
Response: Thank you for your suggestion. If by “review section” you are referring to the discussion section, we have structured it to link the results in a coherent and continuous manner, with the aim of creating a fluid narrative that connects the findings with the relevant literature. All authors agreed that this approach brings clarity to the manuscript. However, if you feel it is necessary, we would appreciate it if you could indicate which specific sections you recommend summarizing or merging, so that we can address your concern more precisely.
Comment 2: In L146 ‘Three irrigation treatments were applied after fruit harvest, between the third and fourth weeks of November’, Why choose to irrigate three times in this time period? If the time is changed to how many days after fruit harvest is more accurate?
Response: Thank you for your observation. We have clarified that the irrigation treatments were applied immediately after fruit harvest, starting on the same day as harvest, and maintained during the period after harvest. This adjustment provides a more accurate description of the timing of the treatments.
Comment 3: How to determine ‘the specific thresholds of stem water potential’, according to a reference or determined in a pre-experiment?
Response: We appreciate your comment. The thresholds of stem water potential were established considering a stress threshold of −1.3 MPa reported by Blanco et al. (2018) for sweet cherry trees. Based on this reference, we defined two levels of deficit: moderate (−1.3 to −1.6 MPa) and severe (−1.6 to −2.0 MPa), with the latter being more restrictive than those previously reported.
Comment 4: Whether uncontrollable rainfall will affect the experimental results?
Response: Thank you for your comment. We acknowledge that rainfall events occurred during the post-harvest period in the last season; however, we do not consider this as affecting the validity of the results. This study was designed to evaluate the strategy under real-world conditions, including environmental variability, which is inherent to production systems. Observing the treatments across three consecutive seasons, some with higher water demand and others with post-harvest periods interrupted by rainfall, provides valuable insight into the residual effect of the irrigation strategy. Additionally, in the last season, the control treatment corresponded to an approximate surplus of 10% of ETc, which was considered in the interpretation of the results.
Comment 5: The data in the table should be added with significance analysis letters.
Response: Thank you for your observation. All tables include their respective ANOVA and Fisher’s LSD analysis, presenting significance letters only when differences among treatments were detected. In cases where no significant differences were found, we included a column with the p-value, indicating “ns” (not significant) or the exact p-value when applicable. Regarding Table 2, the analysis was purely descriptive; therefore, it does not include significant letters or p-values. To avoid confusion, we have now added the note “descriptive analysis” in Table 2.
